# Thia-Michael Reaction under Heterogeneous Catalysis

Giovanna Bosica [1,*] , Roderick Abdilla [1] and Alessio Petrellini [2]

[1] Green Synthetic Organic Chemistry Laboratory, Department of Chemistry, University of Malta, MSD 2080 Msida, Malta

[2] Green Chemistry Group, School of Sciences and Technology, Chemistry Division, University of Camerino, Via Madonna delle Carceri, 62032 Camerino, MC, Italy

* Correspondence: giovanna.bosica@um.edu.mt; Tel.: +356-2340-3074

**Abstract:** Thia-Michael reactions between aliphatic and aromatic thiols and various Michael acceptors were performed under environmentally-friendly solvent-free conditions using Amberlyst® A21 as a recyclable heterogeneous catalyst to efficiently obtain the corresponding adducts in high yields. Ethyl acrylate was the main acceptor used, although others such as acrylamide, linear, and cyclic enones were also utilized successfully. Bifunctional Michael donor, 3-mercaptopropanoic acid, positively furnished the product, albeit in a lower yield and after leaving the reaction to take place for a longer time. The catalyst was easy and safe to handle and successfully recycled for five consecutive cycles.

**Keywords:** thia-Michael reaction; addition; Amberlyst® A21; solvent-free; heterogeneous catalysis; thiols

## 1. Introduction

C-S bonds are remarkably important functional units in organic chemistry [1–3]. The organo-sulfur compounds are found in very different contexts in everyday life, from the biological sphere as amino acids or natural products (penicillin) to the industrial world (polymers in vulcanization processes; surfactants to induce emulsion formation; food additives for flavor enhancement) (Figure 1) [1–3].

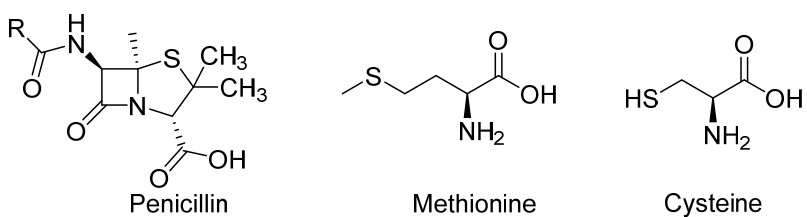

**Figure 1.** Structure of natural sulfur-containing compounds: penicillin (antibiotic), methionine, and cysteine (amino acids).

In the pharmaceutical industry, the importance of sulfur-containing compounds can never be overstated. It suffices to say that in 2011, 25% of the top-200 drugs sold within the US market comprised sulfur in their molecular structure [2]. Historically, sulfonamides have been leading constituents in drugs since the 1930s, and the first antibiotics which were synthesized were similarly of this nature. Back in 1963, the diuretic and antihypertensive cyclothiazide, another sulfur-containing molecule, was introduced in the US market, while a couple of years later, cimetidine was approved by the FDA in 1979 as a drug which inhibits stomach acid production (i.e., it is used to treat acid reflux and heartburn). Sulfur is prevalent in the molecular structure of anticancer agents such as that in Figure 2, MMP3 inhibitors (matrix metalloproteinase-13 involved in the breakdown of collagen II), and

antihypertensive drugs (Diltiazem, a drug which is also used to treat angina as well as decrease high blood pressure) (Figure 2) [3].

**Figure 2.** Pharmaceutically active compounds containing sulfur.

Numerous essential biochemical processes in prokaryotic and eukaryotic cells are tightly linked to sulfur; in fact, many biosynthetic building blocks are activated in the form of coenzyme A thioesters. Highly important cellular detoxification processes depend on sulfur, including, but not limited to, the conversion of cyanide to isothiocyanate and the conjugation of electrophilic toxins to glutathione (a short oligomer composed of glycine, cysteine, and glutamic acid) and related compounds [2]. C–S bonds modifications are also important for stabilizing the transfer RNA (tRNA) structure and for accurate and efficient translation in protein synthesis, a process which occurs in all living cells [2].

Apart from acting as synthons, sulfur-containing compounds can act as auxiliaries in the synthesis of optically active α-hydroxy aldehydes which are themselves highly critical because upon oxidation they result in the formation of compounds such as lactic acid, citric acid, and glycolic acid [2]. Specifically, glycolic acid can be used in creams to promote skin cell regeneration; citric acid is used as a flavoring or chelating agent; and lactic acid is used as its salt in blood fluid mixtures injected in humans after blood loss due to specific trauma. More so, organo-sulfur compounds are very useful in chemo-selective protection of the olefinic double bond in unsaturated carbonyl compounds [2]. One cannot help but mention that sulfur compounds are also used as surfactants which can in turn be utilized for the synthesis of catalysts or used directly in water-based organic reactions [2]. For instance, sodium dodecyl sulfate is needed in the synthesis of the microporous silica SBA (Santa Barbara Amorphous) 15 [2].

Recently, the thia-Michael addition reaction has emerged as one of the most powerful tools for C–S bond formation [1,2] Generally, the 1,4-conjugate addition of a thiol requires the activation of an acceptor olefin by a Lewis acid or deprotonation of thiol by a base [1–3]. In green chemistry terms, it is sustainable because it is 100% atom economic and requires inexpensive starting materials. The first report using an acid catalyst for thia-Michael addition was provided by Thiol et al. [4]. The authors obtained hydrochloride salts of thia-Michael adducts that formed from monosubstituted chalcones and alkylthiols; however, the performed reaction was substrate-selective and was only applicable to chalcones and β-alkylmercaptans. Of late, a base-catalyzed procedure was developed that uses tetrabutylammonium hydroxide in conjunction with green solvents (ethanol or

water) [5]; however, this method requires excessive solvent use during work-up (for solvent extraction), rendering it unsustainable [5].

Alternative developed procedures require the use of acidic catalysts, including the expensive and rare metal catalysts indium bromide and vanadyl acetonate, or heterogeneous solid catalysts, such as silica sulfuric acid which requires the highly corrosive chlorosulfonic acid to prepare it from silica gel [6,7].

Subsequently, other innovative and complex procedures were developed. For instance, the four-component combination of two aldehyde equivalents, thiocarbohydrazide and dialkylacetyledicarboxylate, was performed in the absence of any catalyst or solvent under microwave irradiation at 120 °C for only 10 min. One main drawback of the procedure was that the elevated temperature, which was required to drive the reaction forward, contributed to product degradation soon after formation [8].

In continuation of our previous studies on sustainable hetero-Michael reactions [9,10] in this research, following the 12 principles of green chemistry, a cheap, commercially available, and easily accessible heterogeneous catalyst (Amberlyst® A21) was found to be ideal for the combination of aliphatic/aromatic thiols with common Michael acceptors (Scheme 1) to afford products (11 examples, **3a–k**) in good to excellent yields. The procedure benefits from easy catalyst recovery and minimal solvent use during work-up.

$R^1$ = Ph-, 4-Br-Ph-, HOOCCH$_2$CH$_2$-, Naphthyl-

EWG = -COOEt, -CN, -C(=O)NH$_2$, -C(=O)CH$_3$, -C(=O)CH$_2$CH$_2$-

$R^2$ = H; For dimethyl maleate $R^2$ = EWG = -COOMe

**Scheme 1.** General reaction scheme for thia-Michael addition catalyzed by Amberlyst® A21.

## 2. Materials and Methods

### 2.1. General Information

All the commercially available chemicals were purchased from Aldrich and used without further purification. For the characterization of final products and monitoring of the reactions, the same procedures described in detail in our previous publication were followed in order to obtain FTIR, NMR, and MS and GC spectra [9,10]. Specifically, IR spectra were recorded on an IRAffinity-1 FTIR spectrometer (Shimadzu, Kyoto, Japan) which was previously calibrated at the absorption at 1602 cm$^{-1}$ for a polystyrene strip. Oily products were analyzed as a thin film in between two sodium chloride plates, whereas solid products were dispersed and ground in potassium bromide before pressing into transparent discs using a screwable die. The $^1$H and $^{13}$C-NMR spectra were recorded on an Avance III HD® NMR spectrometer (Bruker, Billerica, MA, USA) equipped with an Ascend 500 11.75 Tesla superconducting magnet and a multinuclear 5 mm PABBO Probe (Bruker). The radio frequency was set at 500.13 MHz for $^1$H spectra, while this was changed to 125.76 MHz for $^{13}$C NMR. Samples (3–5 mg for $^1$H NMR; 20–30 mg for $^{13}$C NMR) were dissolved in deuterated chloroform with TMS as the internal standard. Final spectra processing was performed on MestReNova v12.0.2-20910. Mass spectra were performed using an ACQUITY® TQD system (Waters®) equipped with a tandem quadrupole mass spectrometer and a direct probe that could be directly inserted into the sample and then analyzed. GC analysis for reaction monitoring required the use of a Shimadzu GC-2010 *plus* chromatograph equipped with a flame ionization detector and a HiCap 5 GC column (dimensions: 0.32 × 30 × 0.25) using nitrogen as carrier gas.

## 2.2. General Procedure

The general procedure for the thia-Michel reaction involved stirring the respective thiol (2.5 mmol) and the Michael acceptor (2.5 mmol) in the presence of 0.25 g of dried Amberlyst® A21 catalyst under neat conditions at room temperature for 3 h. The reaction was monitored via both TLCs and/or GC analysis. The catalyst was filtered off by suction and washed appropriately with acetone (approximately 5–10 mL). The filtrate was concentrated by rotary evaporation.

The products were purified by recrystallization from ethanol/hexane (compounds **3d** and **3j**) or by column chromatography using a 9:1, 8:2, 7:3, or 6:4 hexane/ethyl acetate eluant ratio. The TLC plates used for monitoring were composed of silica on PET with fluorescent indicator. Plates were observed under a UV lamp at a wavelength of 254 nm.

## 2.3. Product Identification

**(3a)** *Ethyl 3-(phenylthio)propanoate* [11,12]. Yellow oil. IR (NaCl) ($\nu_{max}$, cm$^{-1}$): 3075, 3059, 2982, 2936, 2874, 1734, 1585, 1483, 1441, 1375, 1346, 1240, 1179, 1096, 1045, 1026, 930, 849, 741, 694. $^1$H NMR (500 MHz, Chloroform-*d*) δ 7.42–7.37 (m, 2H), 7.36–7.29 (m, 2H), 7.24 (tt, *J* = 7.2, 1.3 Hz, 1H), 4.17 (q, *J* = 7.1 Hz, 2H), 3.19 (t, *J* = 7.4 Hz, 2H), 2.64 (t, *J* = 7.4 Hz, 2H), 1.28 (t, *J* = 7.1 Hz, 3H). *m/z* (ES+) = 211.16, 182.05, 165.10, 136.82, 100.91, 73.00.

**(3b)** *3-(phenylthio)propanenitrile* [13,14]. Yellow sticky oil. IR (NaCl) ($\nu_{max}$, cm$^{-1}$): 3076. 3059, 3022, 2970, 2961, 2932, 2872, 2253, 1956, 1882, 1799, 1732, 1585, 1483, 1459, 1421, 1375, 1285, 1240, 1092, 1070, 1049, 1026, 960, 905, 741, 692. $^1$H NMR (500 MHz, Chloroform-*d*) δ 7.47–7.43 (m, 2H), 7.40–7.34 (m, 2H), 7.34–7.30 (m, 1H), 3.15 (t, *J* = 7.3 Hz, 2H), 2.62 (t, *J* = 7.3 Hz, 2H).

**(3c)** *Dimethyl 2-(phenylthio)succinate* [15,16]. Yellow sticky oil. IR (NaCl) ($\nu_{max}$, cm$^{-1}$): 3061, 3003, 2955, 2847, 1728, 1585, 1439, 1410, 1364, 1331, 1304, 1165, 1090, 1070, 1026, 1005, 966, 905, 852, 818, 754, 694. $^1$H NMR (500 MHz, Chloroform-*d*) δ 7.53–7.48 (m, 2H), 7.38–7.34 (m, 3H), 4.05 (dd, *J* = 9.6, 5.7 Hz, 1H), 3.73 (s, 3H), 3.71 (s, 3H), 2.98 (dd, *J* = 17.0, 9.6 Hz, 1H), 2.77 (dd, *J* = 17.0, 5.7 Hz, 1H).

**(3d)** *3-(phenylthio)propenamide* [17]. White solid. MPT = 117 °C. IR (NaCl) ($\nu_{max}$, cm$^{-1}$): 3352, 3182, 3057, 2947, 2918, 2793, 1655, 1481, 1406, 1308, 1283, 1209, 1128, 1092, 1026, 945, 906, 818, 795, 735, 689. $^1$H NMR (CDCl3, 500 MHz) δ 7.42–7.37 (m, 2H), 7.36–7.30 (m, 2H), 7.27–7.21 (m, 1H), 5.46 (d, *J* = 59.7 Hz, 2H), 3.24 (t, *J* = 7.2 Hz, 2H), 2.55 (t, *J* = 7.2 Hz, 2H).

**(3e)** *4-(phenylthio)butan-2-one* [5]. Yellow sticky oil. IR (NaCl) ($\nu_{max}$, cm$^{-1}$): 3065, 3059, 3001, 2932, 1719, 1584, 1483, 1441, 1420, 1364, 1159, 1026, 964, 895, 833, 741, 692. $^1$H NMR (500 MHz, Chloroform-*d*) δ 7.38–7.33 (m, 2H), 7.33–7.29 (m, 2H), 7.24–7.20 (m, 1H), 3.15 (t, *J* = 7.3 Hz, 2H), 2.78 (t, *J* = 7.3 Hz, 2H), 2.16 (s, 3H).

**(3f)** *3-(phenylthio)cyclopentan-1-one* [18]. Yellow sticky oil. IR (NaCl) ($\nu_{max}$, cm$^{-1}$): 3074, 3057, 2968, 2928, 2874, 1745, 1585, 1481, 1441, 1402, 1277, 1244, 1159, 1094, 1026, 980, 943, 897, 743, 694. $^1$H NMR (500 MHz, Chloroform-*d*) δ 7.45–7.42 (m, 2H), 7.37–7.32 (m, 2H), 7.32–7.29 (m, 1H), 3.92 (dq, *J* = 7.4, 6.0 Hz, 1H), 2.67–2.59 (m, 1H), 2.55–2.45 (m, 1H), 2.41–2.33 (m, 1H), 2.31–2.19 (m, 2H), 2.10–1.99 (m, 1H).

**(3g)** *3-((3-ethoxy-3-oxopropyl)thio) propanoic acid*. NOVEL. Yellow oil. IR (NaCl) ($\nu_{max}$, cm$^{-1}$): 3088, 2982, 2932, 2641, 2569, 1734, 1375, 1348, 1250, 1097, 1032, 934, 858, 804. $^1$H NMR (500 MHz, Chloroform-*d*) δ 4.19 (q, *J* = 7.1 Hz, 2H), 2.88–2.81 (m, 4H), 2.70 (t, *J* = 7.1 Hz, 2H), 2.63 (t, *J* = 7.3 Hz, 2H), 1.30 (t, *J* = 7.1 Hz, 3H). $^{13}$C NMR (126 MHz, Chloroform-d) δ 177.43, 171.96, 60.82, 34.81, 34.62, 27.06, 26.73, 14.19. *m/z* (ES+) = 205.24, 189.11, 160.06, 148.79, 133.04, 101.03, 73.00.

**(3h)** *Ethyl 3-(benzylthio)propanoate* [19,20]. Yellow oil. IR (NaCl) ($\nu_{max}$, cm$^{-1}$): 3086, 3063, 3028, 2982, 2932, 2874, 1952, 1884, 1734, 1603, 1456, 1373, 1240, 1179, 1036, 932, 864, 768, 706, 567. $^1$H NMR (500 MHz, Chloroform-*d*) δ 7.37–7.32 (m, 4H), 7.28–7.24 (m, 1H), 4.17 (q, *J* = 7.1 Hz, 2H), 3.76 (s, 2H), 2.71 (t, *J* = 7.1 Hz, 2H), 2.57 (t, *J* = 7.6 Hz, 2H), 1.28 (t, *J* = 7.1 Hz, 3H).

**(3i)** *ethyl 3-((4-bromophenyl)thio)propanoate*. White solid. NOVEL. MPT = 32 °C. IR (NaCl) ($\nu_{max}$, cm$^{-1}$): 3051, 2982, 2928, 2860, 2855, 1736, 1476, 1375, 1346, 1246, 1213, 1182,

1094, 1070, 1009, 930, 812, 741, 706. [1]H NMR (500 MHz, Chloroform-*d*) δ 7.44 (d, *J* = 8.5 Hz, 2H), 7.25 (d, *J* = 8.5 Hz, 1H), 4.17 (q, *J* = 7.2 Hz, 2H), 3.17 (t, *J* = 7.4 Hz, 2H), 2.63 (t, *J* = 7.4 Hz, 2H), 1.28 (t, *J* = 7.1 Hz, 3H). [13]C NMR (126 MHz, Chloroform-*d*) δ 171.56, 134.57, 132.09, 131.60, 120.51, 60.83, 34.31, 29.16, 14.19. *m/z* (ES+) = 291.05, 290.04, 244.99, 243.04, 217.02, 215.00, 202.97, 201.02, 186.97, 101.03, 73.00.

**(3j)** *3-(benzylthio)propenamide* [21,22]. White solid. MPT = 114 °C. IR (NaCl) (ν$_{max}$, cm$^{-1}$): 3350, 3181, 3055, 3028, 2988, 2924, 2843, 2795, 2307, 1946, 1879, 1655, 1414, 1267, 1219, 1069, 1030, 918, 897, 748, 706. [1]H NMR (500 MHz, Chloroform-*d*) δ 7.37–7.32 (m, 4H), 7.30–7.25 (m, 1H), 5.47 (d, *J* = 76.4 Hz, 2H), 3.78 (s, 2H), 2.76 (t, *J* = 7.2 Hz, 2H), 2.44 (t, *J* = 7.2 Hz, 2H).

**(3k)** *ethyl 3-(naphthalen-1-ylthio)propanoate*. NOVEL. Yellow sticky oil. IR (NaCl) (ν$_{max}$, cm$^{-1}$): 3055, 2982, 2934, 2905, 1740, 1734, 1566, 1504, 1373, 1346, 1244, 1179, 1144, 1040, 1026, 976, 930, 804, 793, 772, 667. [1]H NMR (500 MHz, Chloroform-*d*) δ 8.46 (dq, *J* = 8.7, 0.8 Hz, 1H), 7.91–7.86 (m, 1H), 7.80 (d, *J* = 8.2 Hz, 1H), 7.67 (dd, *J* = 7.2, 1.2 Hz, 1H), 7.59 (ddd, *J* = 8.4, 6.8, 1.5 Hz, 1H), 7.54 (ddd, *J* = 8.1, 6.8, 1.4 Hz, 1H), 7.45 (dd, *J* = 8.2, 7.2 Hz, 1H), 4.14 (q, *J* = 7.1 Hz, 2H), 3.25 (s, 2H), 2.63 (t, *J* = 7.4 Hz, 2H), 1.25 (t, *J* = 7.1 Hz, 3H). [13]C NMR (126 MHz, Chloroform-*d*) δ 171.78, 134.08, 133.45, 132.27, 129.90, 128.62, 128.08, 126.61, 126.32, 125.57, 125.26, 60.72, 34.56, 29.64, 14.19. *m/z* (ES+) = 261.18, 215.95, 186.97, 172.95, 101.03, 73.00.

## 3. Results and Discussion

During the initial screening, the reaction between thiophenol (**1a**) and ethyl acrylate (**2a**) was chosen as the model reaction (Table 1), based on the relatively low cost of both reactants. As evidenced, out of the catalysts tested, the best yields were separately obtained with acidic, basic, and neutral alumina, as well as Amberlyst® A21, as catalysts. Positively, the latter could be used at room temperature to afford the product **3a** in an excellent yield of 95% after only 3 h (entry 6, Table 1). In comparison to the reaction involving basic alumina at RT (entry 5, Table 1), the A21-catalyzed reaction required less solvent during the work-up and filtration was easier, considering its physical nature (spherical beads).

**Table 1.** Optimization runs of model reaction between thiophenol (**1a**) and ethyl acrylate (**2a**) to yield product (**3a**).

| Entry [a] | Catalyst [b] | T (°C) | Time (h) | Yield (%) of 3a [c] |
|---|---|---|---|---|
| 1 | Acid alumina | 50 | 2 | 94 |
| 2 | Neutral alumina | 50 | 2 | 95 |
| 3 | No | 50 | 2 | 47 |
| 4 | Basic alumina | 50 | 2 | 95 |
| 5 | Basic alumina | r.t | 3 | 90 |
| 6 | **Amberlyst® A21** | **r.t** | **3** | **95** |
| 7 | Amberlyst® A15 | r.t | 3 | 40 |
| 8 | Montmorillonite K-10 | r.t | 3 | 55 |

Notes: [a] All reactions were carried out on a 2.5 mmol scale using a 1:1 molar ratio of 1a:2a. [b] The amount of catalyst is 0.25 g. [c] Yield of pure isolated product following column chromatography.

In order to demonstrate the generality of our protocol, the optimized reaction conditions were applied to a variety of Michael acceptors (**1**) and donors (**2**) (Figure 3). All reactions were performed on 2.5 mmol scale. Of note, the relatively unreactive Michael acceptor acrylamide still managed to yield product **3d** (99%). Cyclic and linear enones also afforded the products in appreciable yields (products **3e**, **3f**) albeit after a longer reaction time. The main reason for the lower reactivity of cyclopentenone to yield **3f** is potentially caused by the decrease in orientational stability that a 5-membered ring has (taking into account factors such as bond angle strain, 1,3-transannular interactions and gauche interactions). After these results, we decided to also try an aliphatic bifunctional Michael donor in 3-mercaptopropanoic acid. Positively, the product (**3g**) was collected, albeit in a relatively poor yield, after leaving the reaction to take place for 24 h using a 0.5 equivalent excess of ethyl acrylate in order to ensure complete consumption of the thiol. A possible reason for the appreciably lower yield for 3-mercaptopropanoic acid could be that the latter was adsorbing and reacting with the catalyst via its acidic carboxylate moiety. Other aromatic thiols furnished the respective product in good yields (products **3i** and **3k**), as did benzyl mercaptan (products **3h**, **3j**). Product **3k** was furnished at a slightly lower yield compared to the other products potentially due to steric hindrance and also because of the lower reactivity of 1-thionaphthol compared to thiophenol. One needs to keep in mind that as the number of aromatic rings increases, the degree of stabilization is usually larger (more resonance forms) rendering the compound less likely to react. In conjunction, the negative charge density of the nucleophilic sulfur atom could be distributed over two rings, hence making it less likely to attack the electron-acceptor (ethyl acrylate).

Meanwhile, **3j** was also produced at a marginally inferior yield for two possible reasons. Firstly, the acceptor used, acrylamide, is rendered less electron-deficient compared to ethyl acrylate or dimethyl maleate because of the amino group that donates electron density onto the carbonyl group making it less able to itself decrease the charge density from the alkene moiety (by resonance). Secondly, benzyl mercaptan possibly has a higher tendency to undergo oxidation because of the highly reactive methylenic group. Lastly, being aliphatic, the benzylthiolate anion which is formed upon deprotonation is not stabilized by resonance (as is the thiophenolate ion). Consequently, the intermediate is of a higher energy than that created by the attack of thiophenol. As per Hammond's postulate this means that the transition state leading to its formation is also of a higher energy amount. As a result, the activation energy of the process is higher, and the process becomes less kinetically feasible.

### 3.1. The Catalyst—Properties and Recyclability

As already accentuated, Amberlyst® A21 is easily recoverable, and, in fact, it could be recycled and reused effectively for five subsequent runs for the synthesis of product **3a,** as pictorialized in Figure 4.

Amberlyst® A21 is a polymeric material in the form of macroporous beads (490–690 μm diameter range). It is considered to be a free neutral base because it is essentially composed of styrene divinylbenzene units with tertiary *N*,*N*-dimethylamino groups bonded to the matrix (Figure 5). Being in the form of beads, Amberlyst® A21 is practical and safe to use on an industrial level because it cannot result in the formation of harmful airborne particulate matter [23]. Furthermore, it has wide scope of synthetic catalytic applicability, such as in the classical Michael reaction and in the nitro-Mannich reaction where, in conjunction with copper iodide supported on it, it is crucial in the formation of the imine from an aldehyde and an amine which is then attacked by the nitro-compound [24,25].

Mechanistically speaking, the role of Amberlyst®A21 as a catalyst is in all likelihood to be dual in nature. Essentially, considering that it is porous, the internal cavities help to act as microreactors where reactants are more likely to come into close contact, interact and ultimately react. In addition (see Scheme 2), the tertiary amine groups of Amberlyst® A21 help to deprotonate the relatively acidic thiol group to generate a thiolate ion. Thereafter, the latter can nucleophilically attack the β-position of the conjugated ketone/ester to

furnish the product in enolate form. The protonated version of the catalyst is then necessary to donate a proton to the latter mentioned intermediate, form the product and ultimately become basic in nature once again.

**Figure 3.** Substrate scope expansion for the thia-Michael reaction catalyzed by Amberlyst® A21 at room temperature in neat conditions. Notes: [a] Unless otherwise stated, all reactions were performed in neat conditions at RT using a Michael acceptor/donor molar ratio of 1:1 on a 2.5 mmol scale in the presence of dry Amberlyst® A21 (0.25 g). [b] The product was isolated by crystallization from ethanol/hexane. [c] The reaction was left to stir for 24 h at RT. [d] The reaction was left to stir for 24 h at RT and a 0.5 equivalent excess of the Michael acceptor (ethyl acrylate) was used. [e] The reaction was carried out on a 1.25 mmol scale.

Meanwhile, the fact that the acidic catalyst Amberlyst® A15 (which has terminal *p*-toluene sulfonic acid groups) did not furnish the model product in a good yield means that it was probably not acidic enough to activate the carbonyl group (of the Michael acceptor) for attack by the nucleophile. On the other hand, alumina was better possibly because it could act as a Lewis acid by accepting electron density from the carbonyl group (of the Michael acceptor). Nonetheless, as accentuated previously Amberlyst® A21 was preferred owing to its physical nature.

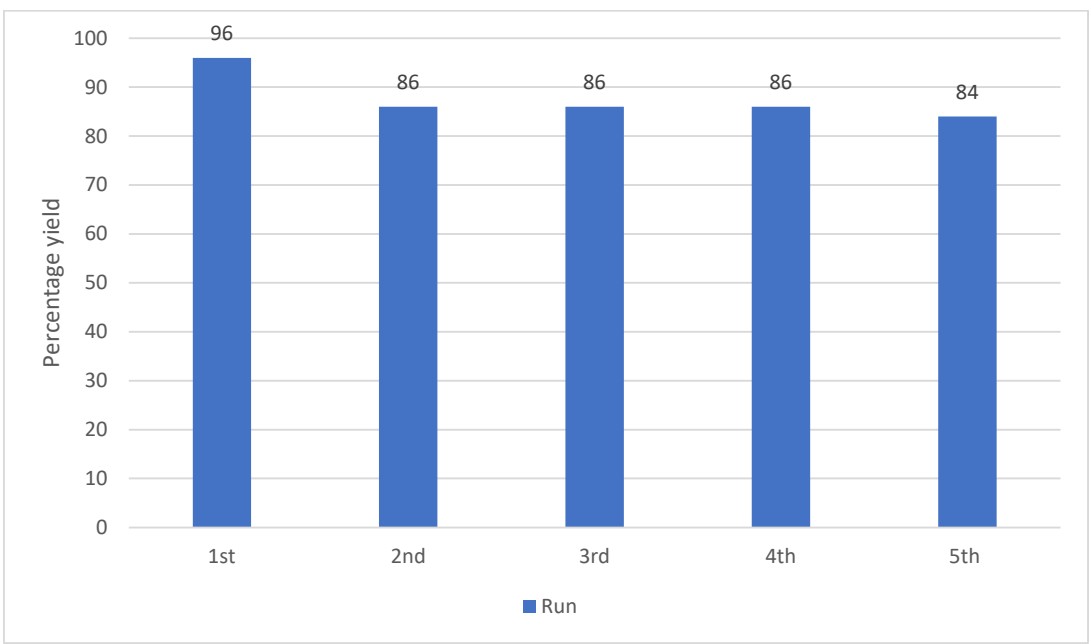

**Figure 4.** Recycling runs for the synthesis of model reaction product **3a**.

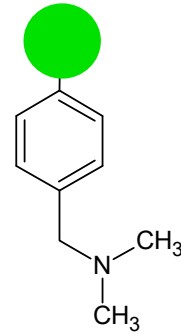

**Figure 5.** Structure of Amberlyst® A21.

Amberlyst® A21 compares very well to other previously reported catalysts in the thia-Michael reaction. The chief aspect of the above-reported system is that the reaction could be performed at room temperature, the catalyst is inexpensive, readily available, could be recycled, and the reaction could be performed in neat conditions with minimal solvent needed for work-up (owing to the physical nature of the catalyst). For the model product **3a**, Table 2 attempts to highlight the main differences between our strategy and those of other scientific groups:

**Table 2.** Comparison of A21 catalyst and other reported catalysts in the synthesis of product **3a**.

| Catalyst Used | Solvent | Temperature/Time | Yield | Reusability | Reference |
|---|---|---|---|---|---|
| 0.5 mol% HClO$_4$-SiO$_2$ | CH$_2$Cl$_2$ | RT/10–15 min | 95% | 4 times (11% yield drop) | [26] |
| 0.5 mol% RuCl$_3$ | PEG 2000 | 50 °C/8 h | 92% | 5 times (4% yield drop) | [27] |
| Amberlyst® A21 | Neat | RT/2 h | 95% | 5 times (12% yield drop) | -<br>(this study) |

**Scheme 2.** The mechanism of the Amberlyst® A21-catalysed thia-Michael addition between thiophenol and ethyl acrylate.

### 3.2. E-Factor and Atom Economy of the Reaction

The thia-Michael reaction is a 100% atom economic reaction because all the atoms in the starting materials are incorporated in the final product.

Meanwhile, the E-factor is defined as the mass of waste produced during the reaction divided by the mass of product which is collected. In order to do so, the solvent used during the reaction (if any) should be taken into account as should the catalyst and the yield of the process. The mass of the waste produced in the process can be calculated by subtracting the mass of the product and recovered catalyst from the mass of the starting materials (the reactants, catalyst and the solvent).

For the model reaction, therefore, the mass of waste produced = [0.275 g (mass of thiophenol) + 0.25 g (mass of ethyl acrylate) + 0.25 g (mass of Amberlyst® A21)] − [0.25 g (mass of recovered catalyst) + 0.504 g (mass of product i.e. 96% of theoretical mass of 0.515 g)] = 0.021 g.

Following the above, the E-factor is equal to:

$$E - factor = \frac{0.021}{0.504} = 0.0416$$

The resulting value is relatively very small which is a positive aspect in light of the green chemistry principles because it means that little to no waste is generated in order to form the product.

### 4. Conclusions

The easily and commercially available, cheap, spherical-bead polymeric resin Amberlyst® A21 was found to be an excellent catalyst in the synthesis of 11 products via the thia-Michael addition of aliphatic/aromatic thiols with various Michael acceptors. All reactions were performed under green, heterogeneous, and solvent-free conditions at room temperature, in the presence of 0.1 g of Amberlyst® A21 per mmol of substrate, and provided good to excellent yields. The catalyst could be recovered easily by simple filtration and reused for up to 5 consecutive times. The reaction had a very low E-factor and is 100% atom economic which makes it highly attractive in green chemistry terms. The fact that a bifunctional

substrate such as 3-mercaptopropanoic acid could also furnish the product opens the door to more functionalized end products that could have pharmaceutical applications.

**Supplementary Materials:** The following supporting information can be downloaded at: https: //www.mdpi.com/article/10.3390/org4010007/s1.

**Author Contributions:** G.B. conceived and designed the experiments; A.P. and R.A. performed the experiments; G.B., A.P. and R.A. analyzed the data; G.B. wrote the paper. All authors have read and agreed to the published version of the manuscript.

**Funding:** This research received no external funding.

**Data Availability Statement:** Experimental procedures and spectroscopic data are provided in the Supporting Information.

**Acknowledgments:** The authors thank the University of Malta for financial and technical support. The authors would also like to thank Duncan Micallef for assistance with the acquisition of the NMR spectra, and Godwin Sammut for MS analyses.

**Conflicts of Interest:** The authors declare no conflict of interest.

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
