# Peer review of "Thia-Michael Reaction under Heterogeneous Catalysis"

_organics, doi:10.3390/org4010007_

Round 1

Reviewer 1 Report

Manuscript presented by Giovanna Bosica et al. shows an experimental study about synthesis of  polycyclic pyridines in thia-Michael addition catalysed by Amberlyst® A21. According to the Authors, the protocol of reaction is consistent with the 12 principles of green chemistry. The topic is very important due to environmentally-friendly aspects of design processes for chemical synthesis.

An already well written and prepared manuscript. Easy to read and follow. Some aspects should be improved. I recommend the article to publish in Organics (MDPI) but first the paper should be corrected. My decision – reconsider after minor revision. Comments to be considered, in order to further improve the manuscript quality:

(1)   Keywords: Please change “aromatic and aliphatic thiols” on “thiols”

(2)   Please add paragraph about properties and safety of “Amberlyst® A21” and its application into similar reaction.

(3)   Scheme I – to better tracking the text I suggest add table with combination of R1, R2 and EWG with the corresponding number of compound, eg. R1=Ph; R2=H; EWG=-COOEt; 3a. I suggest to change also “11 examples” on “3a-k”.

(4)   Helpful to underline the role sulphur organic compounds, their role and synthesis in simple protocol in cycloaddition have been described recently in publications: Eur. J. Org. Chem. 2020, 176–182 (10.1002/ejoc.201901443);Molecules 2021, 26, 5562 (10.3390/molecules26185562). Publications can be helped to improve and diversify the introduction.

(5)   Add information into the manuscript if 3g, 3i and 3k are new (no references).

(6)   In the conclusions, please refer to other synthesis methods of presented compounds. Based on this, please underline advantage the protocol used by the Authors.

(7)   The English correction is necessary. Use British English throughout the manuscript.

I hope the Authors will consider those comments and modify their writings as well as the appearance of the manuscript.

Author Response

(1)   Keywords: Please change “aromatic and aliphatic thiols” on “thiols”

Changed.

(2)   Please add paragraph about properties and safety of “Amberlyst® A21” and its application into similar reaction.

Added towards the end in catalytic recycling section.

(3)   Scheme I – to better tracking the text I suggest add table with combination of R1, R2 and EWG with the corresponding number of compound, eg. R1=Ph; R2=H; EWG=-COOEt; 3a. I suggest to change also “11 examples” on “3a-k”.

Changes made. Table 2 now shows what is the nature of R1, R2 and EWG for each product. However, no such changes were made in Scheme 1 because it would essentially be a repetition of Table 2.

(4)   Helpful to underline the role sulphur organic compounds, their role and synthesis in simple protocol in cycloaddition have been described recently in publications: Eur. J. Org. Chem. 2020, 176–182 (10.1002/ejoc.201901443);Molecules 2021, 26, 5562 (10.3390/molecules26185562). Publications can be helped to improve and diversify the introduction.

We don’t find the cited texts as relevant to our reaction or containing any information that can further accentuate the importance of organo-sulphur compounds.

(5)   Add information into the manuscript if 3g, 3i and 3k are new (no references).

The word NOVEL has been added to show that they are newly synthesized compounds

(6)   In the conclusions, please refer to other synthesis methods of presented compounds. Based on this, please underline advantage the protocol used by the Authors.

Done in the form of a table in the discussion section.

(7)   The English correction is necessary. Use British English throughout the manuscript.

Some changes made to ensure that text is as per British English regulations

Reviewer 2 Report

This Manuscript presented by Giovanna and co-workers describes the Thia-Michael reaction under heterogeneous catalysis. This is a well-known study, but the reusability of the catalyst makes this study interesting. I feel that this paper may deserve to be published in Organics after the following points are revised correctly.

1. Page 1, Abstract, Replace "solventless" with "Solvent-free"

2. Add some more important sulfur-containing molecules in Figure 1

3. The importance of organosulfur compounds needs more discussion

4. Give proper reference to the statement ".......olefinic double bond in unsaturated carbonyl compound.

5. Page 2, Line 1, Replace "cheap" with 'inexpensive'.

6. Page 2, third paragraph, Rewrite the sentence "....required to drive the reaction resulted in some products getting degraded soon after their formation. (.......to drive the reaction may lead to product degradation.)

7. Page 5, Final paragraph, Rewrite the sentence. obtained using acidic,... (The best results are obtained with acidic .......)

8. Page 8, Table 2, correct the typographical error 0,25 g

Author Response

  1. Page 1, Abstract, Replace "solventless" with "Solvent-free"

Corrected

  1. Add some more important sulfur-containing molecules in Figure 1

Added

  1. The importance of organosulfur compounds needs more discussion

Done

  1. Give proper reference to the statement ".......olefinic double bond in unsaturated carbonyl compound.

Reference replaced

  1. Page 2, Line 1, Replace "cheap" with 'inexpensive'.

OK, done.

  1. Page 2, third paragraph, Rewrite the sentence "....required to drive the reaction resulted in some products getting degraded soon after their formation. (.......to drive the reaction may lead to product degradation.)

Reworded

  1. Page 5, Final paragraph, Rewrite the sentence. obtained using acidic,... (The best results are obtained with acidic .......)

Reworded

  1. Page 8, Table 2, correct the typographical error 0,25 g

Done

Reviewer 3 Report

In this work a thia-Michael reactions between aliphatic and aromatic thiols and Michael acceptors have been performed under environmentally-friendly solventless conditions using Amberlyst® A21 as a recyclable (up to 5 times) heterogeneous catalyst to efficiently obtain the corresponding adducts in high yields.

This reviewer thinks that this article deserves to be accepted for publication in organics as there is enough novelty in it only after minor revisions.

1.   It is needed to add a reference: Recently, the thia-Michael addition reaction has emerged as one of the most powerful tools for C–S bond formation. (add reference)

2.     Formatting mistakes

a.     In page 5, error entry 6 and error entry 5.

b.     In page 8, it should be written Figure 2 instead of Figure

c.     In page 7, in the text should be written 3f instead of only 3

3.     Characterization of compounds.

a.     Products 3h, 3j à the integral of the HNMR is not correct in some signals. 

Author Response

  1. It is needed to add a reference: Recently, the thia-Michael addition reaction has emerged as one of the most powerful tools for C–S bond formation. (add reference)

Added

  1. Formatting mistakes

a. In page 5, error entry 6 and error entry 5.

Cannot find mistake

b. In page 8, it should be written Figure 2 instead of Figure

Cannot find mistake

c. In page 7, in the text should be written 3f instead of only 3

Corrected

  1. Characterization of compounds.

a. Products 3h, 3j à the integral of the HNMR is not correct in some signals. 

Corrected. The interference of the solvent signal caused the integration software to miscalculate it.